# Multi-Omics Approaches in Colorectal Cancer Screening and Diagnosis, Recent Updates and Future Perspectives

**DOI:** 10.3390/cancers14225545

**Published:** 2022-11-11

**Authors:** Ihsan Ullah, Le Yang, Feng-Ting Yin, Ye Sun, Xing-Hua Li, Jing Li, Xi-Jun Wang

**Affiliations:** 1National Chinmedomics Research Center, National TCM Key Laboratory of Serum Pharmacochemistry, Heilongjiang University of Chinese Medicine, Harbin 150040, China; 2State Key Laboratory of Dampness Syndrome, The Second Affiliated Hospital of Guangzhou University of Chinese Medicine, Dade Road 111, Guangzhou 510260, China

**Keywords:** colorectal cancer, multi-omics, biomarkers

## Abstract

**Simple Summary:**

Colorectal cancer (CRC) is one of the most prevalent cancers worldwide. Due to the absence of specific early symptoms, most of CRC patients are often diagnosed at late stages. Different screening and diagnostic biomarkers are currently used for risk stratification and early detection of CRC, which might prolong the overall survival. High-throughput technologies have witnessed rapid advancements in the last decade. Consequently, the development of multiple omics technologies, such as genomics, transcriptomics, proteomics, metabolomics, microbiomics, and lipidomics, has been widely applied to develop novel biomarkers that could contribute to the clinical management of CRC. In this paper, we aim to summarize the recent advances and future perspectives in using multi-omics technologies in CRC research, and reveal the potential implications of multi-omics for discovering novel biomarkers and enhancing clinical evaluations.

**Abstract:**

Colorectal cancer (CRC) is common Cancer as well as the third leading cause of mortality around the world; its exact molecular mechanism remains elusive. Although CRC risk is significantly correlated with genetic factors, the pathophysiology of CRC is also influenced by external and internal exposures and their interactions with genetic factors. The field of CRC research has recently benefited from significant advances through Omics technologies for screening biomarkers, including genes, transcripts, proteins, metabolites, microbiome, and lipidome unbiasedly. A promising application of omics technologies could enable new biomarkers to be found for the screening and diagnosis of CRC. Single-omics technologies cannot fully understand the molecular mechanisms of CRC. Therefore, this review article aims to summarize the multi-omics studies of Colorectal cancer, including genomics, transcriptomics, proteomics, microbiomics, metabolomics, and lipidomics that may shed new light on the discovery of novel biomarkers. It can contribute to identifying and validating new CRC biomarkers and better understanding colorectal carcinogenesis. Discovering biomarkers through multi-omics technologies could be difficult but valuable for disease genotyping and phenotyping. That can provide a better knowledge of CRC prognosis, diagnosis, and treatments.

## 1. Introduction

Colorectal cancer (CRC) is the third most common cancer accounting for 10.2% of new cases and 9.2% of Cancer-related mortality, thus accounting for the second most deadly cancer globally [1]. It has been reported that the overall survival rate of metastatic CRC (mCRC) at 5 years over the first examination lowers from 87–90% in stages I–II, and 68–72% in stages III; in stage IV, the rate drops to 11–14% [2]. Most CRC treatment options currently rely on cancer staging, patient performance status, RAS, BRAF, ERBB2, and mismatch repair (MMR) status assessments using tumour samples taken during surgery or core biopsy [3,4]. At present, for patients with mCRC, it is recommended to determine KRAS/NRAS and BRAF mutation status, as well as HER2 amplification and microsatellite instability high (MSI)/mismatch repair (MMR) status (if not performed already) [4]. Recent studies have demonstrated that immune checkpoint inhibitor therapy is effective in treating dMMR/MSI-H mCRC tumours at advanced stages of the disease [5]. The discovery of new molecular biomarkers in CRC and other cancers has begun to follow the approval of tumour-agnostic drugs, including NTRK1-3 translocations and high tumour mutational burdens (TMBs) [6,7]. As opposed to metastatic cancer, there are still no validated biomarkers indicating which patients are more likely to benefit from adjuvant cytotoxic therapy in stage II or III CRC, except for microsatellite instability (MSI) [8]. Additionally, postoperative treatments are often administered following metastatic resections, despite the absence of predictive biomarkers [3,4]. Currently, Colonoscopy, tissue biopsy, and fecal occult blood test (FOBT) are the major techniques used in CRC screening and detection. However, in the case of Colonoscopy or biopsy, these techniques are invasive, causing discomfort for the patient, or in the case of FOBT, they may also have low sensitivity [9,10,11]. Therefore, it is demonstrated that a less invasive test with higher sensitivity is needed in clinical practice.

In particular, high throughput “multi-omics” technologies, including genomics, transcriptomics, proteomics, microbiomics, and metabolomics, provide less or noninvasive approaches for diagnosing CRC. Each method offers a unique advantage for the discovery of novel diagnostic cancer biomarkers, such as Genomics, which is incredibly efficient for evaluating CRC vulnerability and the disease’s genetic risk. However, it has little diagnostic potential since DNA sequences seldom translate directly to phenotype due to epigenetic, post-transcriptional, and post-translational alterations [12]. Transcriptomics and proteomics have great therapeutic potential as they are more closely tied to organisms’ physiological states. Still, their diagnostic power is not as good as that of metabolomics, which enables quick and precise phenotypic characterization of the organism and its metabolic pathways as well as the potential to evaluate how host and gut bacterial metabolites interact, which is a crucial step in the CRC progression [13]. Additionally, a number of recent research have shown that the gut microbial community and microbial metabolites play a crucial role in the emergence of CRC [14,15]. Recent years have seen the emergence of lipidomics as a research tool and a multi-omics technology that holds great promise. As a result, this tool has been demonstrated to be useful for both the quantification of cellular lipids and their characterization. This is not only for disease diagnosis but also for other mechanistic studies [16,17]. The mechanism of CRC initiation and progression has remained largely enigmatic despite the discovery of more diagnostic methods and potential therapies; many challenges remain unresolved due to the lack of new biomarkers and the heterogeneity of tumours. After the completion of the human genome project, omics science has revolutionized CRC research [18]. In order to enable personalized medicine and to define CRC treatment, the identification of novel biomarkers has become an essential part of molecular diagnosis and treatment [19]. The use of new biomarkers in clinical practice is still challenging despite developments in the molecular analysis [20]. However, genomic advances have made significant contributions to understanding cancer biology over the past few years [21]. In oncology, the structure and functions of the genome, as well as mechanisms governing genes’ expression, have been extensively investigated since the completion of the Human Genome Project and the development of next-generation sequencing (NGS) techniques [21,22]. A constantly expanding understanding of genomic hallmarks of malignant transformation provides a new perspective on pathogenesis and targeted treatment of particular tumours [23,24]. Genomics, transcriptomics, proteomics, metabolomics, microbiomics, and lipidomics, make a significant contribution to a fundamental change toward a multiparametric, innovative, immunological, and stromal model, which helps us to understand how CRC develops and categorizes it into various molecular subtypes for clinical diagnosis as well as the emergence of new biomarkers and therapeutic strategies [25], (Figure 1).

The aim of this review is to summarize the recent developments in multiple multi-omics technologies in the exploration of CRC biomarkers signatures via genomics, transcriptomics, proteomics, microbiomics, metabolomics, and lipidomics. These promising multi-omics base CRC biomarkers could be useful for clinical research.

## 2. Genomics of CRC

Genomic science comprises the study of an individual’s entire set of DNA (including all of their genes) [26]. An individual’s genomes are a comprehensive collection of information that enables them to grow and develop [27]. Using genomic analysis, researchers may better understand gene interactions, environmental effects, and how several conditions, such as cancer and diabetes, develop [28]. The development of these new approaches may facilitate disease diagnosis, treatment, and prevention [29]. During carcinogenesis, genetic and epigenetic changes occur that contribute to the identification of ideal biomarkers of CRC [30]. There is growing evidence that genetic changes play a key role in tumorigenesis. Due to this, genomics is becoming a powerful tool for finding genetic markers that can be used to diagnose and prognosis cancer, as well as improving our understanding of the disease. High-throughput next-generation sequencing is a genomic technique for sequencing an organism’s DNA [31]. Multiple- biomarker panels are usually more sensitive than single biomarkers, as demonstrated by many research studies over the last few years [32]. For illustration, Ghatak S et al. used differential gene expression analysis in five independent in silico CRC cohorts and immunohistochemistry in one clinical cohort to validate their results. The authors developed a novel biomarker for early diagnosis and prognosis of cancer based on a five-panel gene signature [33]. All five in silico datasets showed that four genes (PTGS2, BDNF, CTNNB1, and GSK3B) were highly upregulated. One gene (HPGD) was substantially downregulated in primary tumour tissues compared to neighbouring normal tissues. Based on independent clinical validation cohorts, this five-gene signature was significantly associated with poor overall survival (AUC = 0.82) among colon cancer patients.

An epigenetic change happens when modified nucleotide sequences in the genome appear to be altered beyond their original form [34]. Gene expression is regulated by epigenetic mechanisms such as DNA methylation, histone modification, and nucleosome positioning Inhibitions in these regulatory processes promote malignant transformation by impairing gene function [35,36]. There is abnormal methylation of the CpG promoter during CRC, which leads to promoter hypermethylation in the promoters of tumour suppressor genes and the silencing of the transcriptional activity of DNA repair genes, which is accompanied by a loss of methylation (hypomethylation) that contributes to oncogene activation, chromosome instability, and microsatellite instability [37,38]. There is evidence that CRC biomarkers such as methylation in cfDNA and CTCs may be useful for the noninvasive diagnosis of CRC [39,40]. In addition, the epigenetic modification of 5-methylcytosine (5mC) has been associated with the emergence of several disorders, including CRC. An increasing number of studies suggest that 5mC can be used in diagnoses and prognosis of colorectal Cancer [41,42,43,44,45,46,47]. Furthermore, members of the ten-eleven translocation family catalyze the production of 5-hydroxymethylcytosine (5hmC), a persistent byproduct of DNA epigenetic regulation. The change of 5hmC, a new epigenetic biomarker, is linked to several disorders, particularly Cancer [48,49,50,51,52,53]. There is evidence that 5hmC plays an important role in the progression of CRC [47,53]. However, it has rarely been studied as a potential diagnostic marker for the early detection of CRC. The potential genomics biomarkers are shown in Table 1.

## 3. Transcriptomics of CRC

A transcriptomic study analyzes an organism’s entire RNA content. Transcription represents an overview of the cell’s activity at a particular moment due to the information in DNA [64]. In recent years, transcriptomics has made unprecedented progress in molecular genetics [21,65]. At certain developmental stages and under certain physiological or pathological conditions, transcriptomes represent all RNA molecules produced in a cell from the genome [66,67]. It consists of protein-coding RNAs (pcRNAs), also known as messenger RNAs (mRNAs), and non-coding RNAs (ncRNAs), of which each molecule exhibits a wide range of cellular functions and responses to external stimuli [68,69,70,71]. As a result of epigenetic changes and genomic instability, transcriptome changes may occur in CRC. In CRC, ncRNAs play an important role in angiogenesis, migration, differentiation, and apoptosis. Therefore, the study of ncRNAs is one of the most prominent areas of RNA research. Numerous studies have provided evidence that ncRNA expression is abnormal in CRCs. A study of ncRNA stability in stool, plasma, and serum may provide new possibilities for developing new methods of detecting ncRNAs, and it has been demonstrated that among ncRNAs, microRNAs have significant impacts on CRC [72,73]. By using next-generation sequencing, deep sequencing of CRC tumours was performed to examine the miRNA transcriptome results demonstrating that CRC patients had increased levels of miRNA-615-3p and miRNA-10b-5p expression in both the right and left side of the colons correspondingly. Additionally, five miRNAs were found to be significantly elevated in CRC patients in the study, including miR-143-3p, miR-22-3p, miR-192-5p, miR-21-5p and miR-10a-5p [74]. Several studies have been conducted to identify novel miRNA as biomarkers, and several studies have demonstrated an important role for miR-92a and miR-429 in CRC pathogenesis [75,76]. In contrast, several miRNA molecules have been demonstrated to have significant diagnostic value for advanced neoplasia, including miR-17-92, miR-135, miR-143, and miR-145.

Moreover, a recent study improved and facilitated exosome-miRNA identification in blood using SHERLOCK-based miRNA detection. It revealed that miR-23a, miR 126, miR-940 and miR-1290, are the best good prognostic indicators for the initial stages of CRC [77]. Several miRNAs including MiR-192a, miR-29a, miR-19a-3p, miR-92a-3p, miR-125b, miR-422a and miR-223-3p, have been considering significant CRC marker. However, miR-21 has been studied extensively for diagnosing CRC [78]. In another study, miR-429 was found to reside at the centre of a miRNA-target gene network, indicating that it plays a critical role in cancer development. The miRNA samples from 28 patients with CRC markedly showed an increase in miR-32 levels. It has been determined that miR-32 expression and CRC lymphatic invasion and metastasis are correlated by the cancer genome atlas (TCGA), and a negative association was also observed between miR-32 and bone morphogenetic protein 5 (BMP5) [79]. By inhibiting EPST11 activation, BMP5 acts as a tumour suppressor. Alteration in BMP5 levels triggers the epithelial-mesenchymal transition, which stimulates tumorigenesis. Sporadic CRC tissues show a positive correlation between BMP5 expression and E-cadherin expression. Yamada et al. identified four lncRNAs, including CRCAL-1, CRCAL-2, CRCAL-3, and CRCAL-4, which differ in expression among normal mucosa and CRC patients through RNA sequencing [80]. The findings of this research highlight the implication of RNA-Seq for identifying new lncRNAs in colorectal cancer. CRC tissue also showed downregulation of NONHSAT074176.2 under GO and KEGG analysis, which may serve as a valuable diagnostic biomarker [81].

A fusion transcript (FT) is a chimeric RNA that comes from a single gene product or the trans-splicing of a transcript made by two gene products. FTs play an important role in the regulation of cancerous cells. It has been reported that transcripts from COMMD10-AP3S1, CTB-35F21.1-PSD2, and AKAP13-PDE8A are the most frequently reported transcripts in CRC. According to another study, higher levels of NFATC3-PLA2G15 fusion transcript were detected in 19 pairs of CRC tumours and adjacent normal tissue samples. As a result of the knockdown of NFATC3-PLA2G15, invasion and proliferation are inhibited in cancer cells, suggesting that NFATC3-PLA2G15 FTs may influence CRC progression; these impact findings show that this fusion transcript can serve as a novel biomarker for CRC [82] TGFRN-NOTCH2 fusion transcripts were the only transcripts detected in CRC and adjacent normal tissues from deep transcriptome sequencing. RT-PCR analysis confirmed the findings, suggesting that PTGFRN-NOTCH2 may be an FT gene in CRCs and may serve as a potential biomarker [83]. 

Furthermore, single-cell RNA sequencing (scRNA-seq) assesses the transcriptomic status of specific populations of single cells compared with RNA sequencing (RNA-seq) in which transcript levels are measured across different cell types [84,85]. In microdroplets and microwells, thousands of single cells can be simultaneously barcoded and handled at the same time [84]. Several technologies have been developed that measure mRNAs that are isolated from a single cell, including Quartz-Seq, Smart-seq, Smart-seq2, and CEL-seq [84,86,87]. These different types of mRNA sequencing technologies with distinct purposes. Smart-seq, for example, detects full-length transcripts. During Quartz-Seq, samples are analyzed and pooled according to the 30 end of transcripts and the CEL-Seq barcodes before linear amplification of mRNA [86]. In a recent study, the transcriptional profiles of 371,223 cells from colorectal cancer and neighbouring normal tissues were taken from 28 tumours with mismatch repair proficiency, and 34 tumours with mismatch repair deficient [88]. a significant finding of this study is that there is a structured arrangement of T cells within a tumour. In summary, the authors have provided a large number of individuals with colorectal cancer with datasets that contain information about cellular states, gene networks, and tumour transformations [88,89]. The results of scRNA-seq studies are promising because, for each cell type in a tumour, alterations may be associated with patient characteristics, diagnostic methods, therapeutic approaches, and prognosis. In the near future, scRNA-seq could be used clinically to develop customized treatment regimens for each patient based on their genetic information [90]. The potential transcriptomics biomarker is shown in Table 2.

## 4. Proteomics of CRC

Proteins regulate many biological processes, and gene mutations could alter their expression. As well as serving as a source of potential biomarkers, the proteome is also the functional translation of the genome. Compared to the normal proteome, cancer proteome biomarkers are up- or downregulated; Thus, researchers have recently focused their attention on identifying differences between cancerous and normal cells in terms of their expression characteristics. To develop new classification tools for CRC, diagnostic, prognostic, and predictive biomarkers must be developed to detect proteins involved in its development and progression and observe the effects of protein perturbations and modifications. 

Many proteomic techniques have been employed in order to find putative diagnostic biomarkers. According to Ghazanfar et al. [102], protein expression in fresh freeze samples of colorectal cancer tissue (12 individuals) was analyzed using gel electrophoresis in combination with mass spectrometry, demonstrating that number of proteins has been upregulated in colorectal Cancer. These include actin beta-like 2 (ACTBL2). Another study by Hao et al. [103], using high-resolution Fourier transform mass spectrometry, revealed that colorectal tumour tissue overexpressed dipeptidase 1 (DPEP1) Based on the examination of 22 pairs of normal tissues adjacent to cancerous tissue. Yamamoto and colleagues used a global proteomic approach to study formalin-fixed and paraffin-embosted (FFPE) CRC tissue with liquid chromatography (LC)/mass spectrometry (MS). They found a higher concentration of cyclophilin A, annexin A2, and aldolase A in cancerous tissues versus non-cancer tissues [104]. Similarly, in another study, fibroblasts associated with cancer progression were identified from human and mouse tissue. As a result of this study, it has been demonstrated that the proteins LTBP2, OLFML3, CDH11, CDH11, CALU, and FSTL1 play an important role in the migration and invasion of CRCs and have been implicated as stromal biomarkers [105].

Among the potential biomarkers of colorectal Cancer, blood-based markers are some of the most promising for performing early detection and surveillance of CRC because obtaining the specimens is relatively easy and noninvasive with minimal risk [106,107]. A targeted liquid chromatography-tandem mass spectrometry analysis was performed on 213 healthy subjects and 50 colorectal cancer patients by Ivancic et al. [108]. This study identified five proteins, including inter-alpha-trypsin inhibitor heavy-chain family member 4, leucine-rich alpha-2-glycoprotein 1, EGFR, hemopexin, and superoxide dismutase 3, that play a significant role in detecting CRC with 89% specificity and over 70%sensitivity. Furthermore, A protein panel for early detection of CRC was discovered by Bhardwaj et al. [109], by using liquid chromatography/multiple reaction monitoring-mass spectrometry in plasma samples from 96 CRC patients, and 94 controls, using a blood-based profile of five markers, osteopontin, serum paraoxonase lactonase 3, transferrin receptor protein 1, mannan-binding lectin serine protease 1, and amphiregulin. Demonstrated promising performance in screening for colorectal Cancer. Additionally, a number of members of the Serpin family including SERPINC1 (antithrombin-3, AT-III), SERPINA3 (alpha-1 antichymotrypsin, AACT), and SERPINA1 (alpha-1 antitrypsin, A1AT), have been identified as potential biomarkers for colorectal carcinoma and adenomatous polyps by using multiplexed quantification isobaric tags for absolute and relative quantitation (iTRAQ), [110]. The importance of CC chemokines (CCL15, CCL4 and CCL2) has also been assessed in CRC however further research is needed for their utility as diagnostic and clinical markers [111].

Numerous LC-MS-based research has been conducted demonstrating different CRC biomarkers. For instance, Quesada-Calvo et al. [112] suggested KNG1, Sec24C, and OLFM4 as diagnostic biomarkers out of 561 proteins with different expression levels. One other study demonstrated that ACTBL2, Annexin A2, Aldose A, DPEP1, and cyclophilin A could also serve as a biomarker for the early detection and treatment of CRC and provide new therapeutic targets [102,103,104]. As a biomarker source, circulating proteins are widely accepted as a better diagnostic tool for many diseases, particularly CRC [113]. Western blot (WB) and ELISA verification studies demonstrate that MRC1 and S100A9 are higher in CRC patients’ serum compared to healthy individuals [114]. Furthermore, Ivancic et al. demonstrated that serum samples containing LRG1, EGFR, ITIH4, HPX, and SOD3 could detect CRC with 89% specificity and 70% sensitivity. According to these findings, GC, CRP, CD44, and ITIH3 proteins may be able to differentiate CRC depending on its stage [108]. Additionally, Bhardwaj and colleagues [109] showed five protein signatures, including MASP1, AREG, PON3, TR, and OPN, compared with FDA-approved biomarkers derived from blood superior diagnostic performance. CXCL-1 (C-X-C motif ligand 1) and CXCL-8 (C-X-C motif ligand 8) and their receptors have also demonstrated a potential role as biomarkers for CRC prognosis and diagnosis [115], Pczek S et al. conducted a study in which increased levels of CXCL-8 were found in CRC patients when compared to normal subjects. The findings of their research revealed enhanced versatility of CXCL-8 as compared to CEA in CRC diagnosis [116]. The potential proteomics biomarkers are shown in Table 3.

## 5. Microbiomics of CRC

Microbiomics is an emerging field of omics technologies that examines a symbiotic or pathological relationship between microbial communities [127]. Many microorganisms exist in the human microbiota (microbiome), such as bacteria, viruses, fungi, etc. [128,129,130,131,132]. An individual’s gut microbiome is composed of microorganisms and their genetic materials. Over 3 million genes exist in the gastrointestinal tract, which is 150 times more than the human genome. In the gastrointestinal tract, 1013 to 1014 different microorganisms live, and over 30 million genes exist. Approximately 7000 different strains of bacteria comprise the gut microbiome in adults [133]. Gut microbiome signatures in CRC were studied using different approaches by various researchers. Various methods enrich 16S rRNA for variable regions in stool DNA, from amplifying and sequencing the V1, V2, and V4 regions to shot-gun metagenomic sequencing. Here are various methods to enrich 16S rRNA for variable regions in stool DNA, from amplification and sequencing of the V1, V2, and V4 regions to shot-gun metagenomic sequencing [134,135,136]. Various qPCR methods have been used to quantitate the abundance of target microbial genes in samples of interest [137,138,139]. 

Gut microbiomes have shown a significant role in the treatment of CRC; As an example, the gut microbiome may be able to be used for screening, diagnosis, prediction and/or predictive biomarkers. Alternatively, it might be a changeable factor affecting systemic CRC treatment efficacy or prevention [127,140]. The gut microbiota is a screening marker among asymptomatic individuals with high-risk adenomas or CRC. The Fusobacterium nucleatum bacteria, for example, can be examined in faecal samples from patients with adenomas and CRC to serve as a screening biomarker. Detecting and screening for CRC early may also be possible based on metabolic markers and genotoxic metabolites of specific strains [139].

Recent research published in a nature journal examined 33 cancer patients’ blood and tissue samples. It revealed that the blood contained specific gut-derived pathogenic bacterial DNA that may be used to distinguish various types of tumours [141]. Therefore, the authors concluded that further research should be undertaken on this possible microbiome-based tumour diagnosis tool. In addition, the study of pathogenic bacteria (intestinal flora), and their metabolites have been linked to CRC, and the correlation analysis of gut-microbiome and metabolomics have shown a promising role in CRC prevention, treatment and diagnosis [142]. This common gastrointestinal malignancy has also become a hotspot of research in recent years [143,144]. Chen F et al. investigated the macro genomic and metabolomic compositions of serum collected from normal patients, colorectal adenomas, and CRC patients. A total of 885 differential metabolites were found in the serum associated with intestinal bacteria. This led to the Identification of eight serum metabolites that were reproducible and were used to develop categorical diagnostic models for healthy/colorectal adenoma (AUC = 0.84) and healthy/colorectal Cancer (AUC = 0.93) [145].

Some common metabolites of intestinal bacteria in the blood, bile acids, such as short-chain fatty acids and oxotrimethylamine, could be biomarkers for early CRC detection [146,147,148]. According to research by Huang Y et al. [149], Fusobacterium nucleatum plays a major role in promoting colorectal carcinogenesis by increasing tumour-associated metabolites, including 12a hydroxy3oxycholic acid and phosphorylcholine in the serum. Another area of active research is the discovery of biomarkers from microbial metabolomes, as some metabolites derived from the microbiota are associated with colorectal cancer. Microbial metabolites have been identified in several studies as potential biomarkers for CRC; for example, using GC-MS, an analysis of stool metabolites was conducted for CRC patients using a GC-MS technique with the result that there was a higher concentration of acetate and a lower concentration of butyrate and ursodeoxycholic, acid (UDCA) in their stool [150]. Another GC-MS metabolomic study was conducted in CRC tissue in which 19 differentiating metabolites were identified, along with pathway enrichment analyses that demonstrated that CRC patients exhibit a significant disruption of several metabolism pathways, including short-chain fatty acid metabolism, secondary bile acid metabolism, and carbohydrate metabolism [151]. Using NMR, a combined examination of tumour tissue and feces revealed a decrease in butyrate levels in patients with CRC; Fecal and tissue samples had AUCs of 0.692 and 0.717, respectively for diagnosing CRC from normal subjects, An AUC of 0.843 was reported for fecal acetate, which was the strongest indicator of diagnostic performance [152]. According to an MS-based metabolomic analysis in CRC cohorts, polyamine-based metabolites also showed a significant upregulation (N1-acetylspermidine, citrulline, arginine and ornithine) [153]. Integrating microbiome and metabolome data has demonstrated that fecal abundances of microbial-associated polyamines (cadaverine and putrescine) may play a role in colorectal cancer diagnosis [154]. CRC screening can benefit from metabolic markers, as demonstrated in these examples. The potential microbiomics and metabolomics biomarkers are shown in the Table 4.

## 6. Metabolomics of CRC

Metabolomics is a new research area in the omics arena. Refers to an in-depth investigation of low molecular weight substances formed by metabolism in a biological fluid, including metabolic substrates and products, small peptides, lipids, vitamins, and other protein cofactors. In biomarker discovery, metabolomics is one of the fast-growing fields [164,165]. Furthermore, unlike genomics, transcriptomics, and proteomics, it represents the connections between genes and the environment, which allows it to be more precise in describing multifactorial diseases [164,166]. Many biological specimens can be used for metabolomics, most of which can be obtained using noninvasive techniques. Although biomarkers and metabolites can vary from study to study and even between specimens and colorectal cancer levels, they remain useful for diagnosing colorectal cancer [167]. Targeted metabolomics involves quantifying the metabolites linked with particular pathways associated with a specific state of disease [168]. In contrast, the untargeted approach was used in many samples and did not undergo any bias; it often measures as many metabolites as possible [169]. Due to its unique insight into disease origin and development processes, the metabolome remains a key component of disease research. Metabolomics may provide valuable information about the pathology of CRC, identify predictive biomarkers, and evaluate the severity of the disease by examining the metabolomic fingerprint in detail [170]. The metabolomics approach based on urine metabolites can be used to identify cancer biomarkers to distinguish patients with early-stage and advanced-stage colorectal cancer [171]. Lactosylceramide has also been identified as a key metabolite distinguishing Crohn’s disease from ulcerative colitis in untargeted metabolomics [172].

Several metabolomic research studies have been carried out in a small cohort of colorectal cancer patients using several biological samples such as blood, urine, stool, and tissue [173,174]. A comparison was made between metabolic profiles of healthy individuals, as well as of individuals with benign polypoid pathology [175] employing nuclear magnetic resonance (NMR) or gas- and liquid-chromatography coupled to mass spectrometry (GC-MS, LC-MS) as analytical tools. Several studies have shown a negative correlation between stool and urine metabolites in patients with advanced colon cancer. The study’s authors conducted a comparison of plasma, stool, and urine metabolic profiles [176]. There have been studies conducted that identified 154 different metabolites, including those that are produced during the tricarboxylic acid (TCA) cycle, urea cycle, polyamine pathways, glycolysis and amino acids, among others. With the progression of Cancer, the concentrations of these metabolites increased, with the greatest difference found in stage IV. Moreover, polyps and CRC samples were discriminated by metabolite analysis [177]. Ning et al. carried out a research study that revealed 11 upregulated and four downregulated metabolites in urine samples collected from CRC patients and healthy subjects, as shown in Table 4. Patients with CRC who were examined for pathways involved in these metabolites showed increased glycolysis, and amino acid metabolism while showing a decrease in lipid metabolism [163].

Another research has been conducted; they studied the relationships between metabolites and health status in healthy individuals and CRC patients using GC-MS analysis based on a metabolomics-based approach. This study identified several polyamines (putrescine, cadaverine) as potential biomarkers for cancer prognosis [154]. By observing metabolomic alterations in patients with CRC, another study utilizing gas chromatography-mass spectrometry (GC-MS) found that stool fatty acids, particularly increased oleic acid, may be used to screen CRC [161]. UHPLC-MS analysis of stool samples from CRC patients revealed different sphingolipid and cholesteryl esters levels [134]. A recent study of CRC tissues and stools conducted through the proton nuclear magnetic resonance (1H NMR) technology showed that butyrate was downregulated in CRC tissue and stools. At the same time, alanine, lactate, glutamate, and succinate were upregulated [152]. As metabolomics have been made a great contribution to drug discovery, UPLC-MS base metabolites biomarkers from natural compounds have also played an essential role in disease treatments [178], for instance our recent pharmacodynamic metabolomics base study using mice serum revealed that flavonoids and anthraquinones have a role in CRC treatment [179]. A combination of several multi-omics technologies could provide a powerful strategy for making valid conclusions about biomarker signatures for Colorectal Cancer. So far, no single Omics technology offers enough information to demonstrate the detailed molecular mechanism and validation of biomarker signatures.

## 7. Lipidomics of CRC

The field of lipidomics is one of the newest branches of multi-omics technologies. With the help of various analytical techniques, this technology can classify and analyze almost all cellular lipids, to understand their role and characteristics within biological systems. It has been studied on a larger scale for lipid species molecules. Several kinds of disease-specific biomarkers have been found through lipidomics, and the lipid species are linked to disease severity [180,181]. Regarding CRC, A very recent study, by Zaytseva et al. suggested that fatty acid metabolism might be used as a strong predisposition in CRC. It emphasized the significance of targeting lipid dysregulation in future therapeutic strategies [182]. Many studies have been conducted to examine the complex lipid profile of serum tissue samples; Consequently, a specific CRC lipidome has been the subject of ongoing discussions that may have implications for clinical treatment. The elevated levels of VLCFA (Very Long-chain Fatty acids) and lower levels of LCFA (long-chain fatty acids) have been observed in CRC patients’ serum. It was explained that ELOVLs (Elongation of Very Long-chain Fatty acids Protein) may increase VLCFA elongation by increasing saturated or monosaturated fatty acids [183,184]. Based on LC-MS analysis, it was found that saturated triacylglycerols accounted for the majority of perturbations that occurred in CRC progression. The authors attributed these perturbations to LD (lipid droplet) accumulation [185]. According to another study, glycolipids, glycerophosphocholine, and acylcarnitines serum concentrations decreased in CRC patients [186]. In a recent study, Ecker et al. also found an independent prognostic marker in triglyceride lipidomic tissue signatures capable of discriminating against patients and can be used as a prognostic indicator. Through quantitative lipidomics analysis, the author demonstrated altered levels of glycerol, glycerophospholipids and sphingolipids in matched tumour samples. It has been shown that glycerol and sphingolipids can discriminate among patients with distinct mismatch repair proficient and deficient statuses, oncogenic mutations (KRAS/BRAF), or grading [187]. Several diseases have multidimensions and networks of lipid molecules fused with genes and proteins in the molecular mechanism. Lipidomics platforms can be used to analyze and characterize these compounds. Furthermore, common and traditional disease diagnoses can be more difficult to identify therapeutic targets. However, lipidomics technology offers the possibility of easier diagnosis for certain types of diseases. A diagnosis of the disease can be made by lipidomics based on existing biomarkers as therapeutic agents [188,189]. It is also possible to study various protein-lipid interactions, lipid-lipid interactions, and lipid-gene interactions, enabling the development of better diagnostic procedures for advanced diseases. A lipidomics approach has been considered better than traditional approaches for disease investigation because it provides an understanding of systemic metabolisms and their mechanisms and precisely identifies therapeutic targets and diagnostic biomarkers [190].

## 8. Future Perspectives and Conclusions

It is well acknowledged that early cancer diagnosis would enhance patient prognosis and provide a greater knowledge of the disease, decrease mortality, and increase patient satisfaction. There has been a significant advancement in identifying new biomarkers in recent years, paving the way for a more personalized approach to the clinical diagnosis and treatment of CRC [19]. Several DNA, RNA, and protein-based cancer biomarkers have been developed recently through high-throughput research in cancer biomarkers that can be discovered from readily available biological samples such as blood, serum, urine, stool, and tissues. Technological advancements have improved the sensitivity and specificity of cancer-specific biomarkers in CRCs. However, traditional biomarkers in clinical practice do not have high specificities and sensitivity. Therefore, in order to develop an accurate and clinically useful test, it is recommended to discover multiple biomarker panels instead of a single biomarker. By identifying prospective new therapeutic intervention targets that might contribute to the diagnosis of CRC, it is possible to develop an alternative to conventional methods of early detection of cancer.

With the recent developments in high-throughput sequencing technologies, Increasingly, cancer researchers are relying on “multi-omics” data sets. Multi-omics combines a range of omics data sets, including genomics, proteomics, transcriptomics, metabolomics, and microbiomes, for analysis [191,192]. By combining quantitative analyses of multi-omics data and clinical features, we can get insight into alterations at the molecular level and gain a more comprehensive, systemic comprehension of various biological pathways [192,193]. By integrating multi-omics approaches, we can simultaneously uncover how information flows between different levels of omics. It will, therefore, help us to bridge and close the gap between genotypes and phenotypes data. With the advent of this Technology, colorectal cancer can be diagnosed, prognosed, treated, and prevented with greater accuracy in the future. Due to the huge amount of data available, multi-omics and big data analytics are required to interconnect all available information. In particular, integrating patient demographic, genomic, transcriptomic, proteomic, metabolomic, lipidomic, and microbiota data could assist in developing new biomarkers discovery and clinical outcomes prediction.

Another emerging field of multi-omics technologies is microbiomics which offers a non-traditional tool with potential applications in more significant comprehension of tumour biology. Identifying microbial metabolites correlated with the development of colorectal tumours has significant implications for identifying new treatment targets and possible biomarkers for disease screening [194]. There has been significant progress in recently using intestinal bacteria and their metabolites as early detection markers for CRC. The association between CRC and gut bacteria and their metabolites has received much attention recently. Moreover, the gut microbiota’s microbial metabolite composition is frequently renewed and changes depending on the diet, making it more amenable to therapeutic intervention in developing CRC. New paradigms in CRC diagnosis, prevention, and treatment will be provided by elucidating the role of microbial metabolites [15]. CRC prevention, comprehensive treatment planning, and minimizing adverse effects of treatment will be significantly impacted by the gut microbiome in the near future. Individuals’ gut microbiomes vary according to their geographic location, ethnicity, dietary habits, and lifestyles. In the future, clinical research will need to include several factors that contribute to the microbiota of patients, including geography, race, sex, and diet, as well as how systemic cancer treatment affects the microbiome, especially chemotherapy and immunotherapy [195]. 

In recent years, lipidomics has been actively used in the research community and regarded as a cutting-edge example of multi-Omics Technology. Particularly useful in analyzing the structure and function of lipid molecules to analyze changes in their dynamic composition during certain pathological and physiological changes. The alterations in lipid metabolism have also been linked to several kinds of cancer development and progression. Despite this, there is a limited understanding of the metabolic changes of lipids in cancer due to their structural diversity and characteristics, distinct from those of other biomolecules. Several analytical tools in cancer research have been used in lipidomic analysis to determine lipid composition at a large scale. Additionally, in cohort studies, glycero-, glycerophospho-, and sphingolipid levels have been significantly changed between tumours and normal tissue. A marked difference between cancerous and non-diseased tissue in sphingomyelin and triacylglycerol (TG) species [187]. Recent research demonstrated that GZMs (granzymes) proteins have a significant role in carcinogenesis, their role as new biomarkers for CRC prognosis and diagnosis will need further exploration [196]. Furthermore, it is imperative to emphasize the importance of lipidomics and proteomics research for discovering novel biomarkers and diagnosing CRCs. Integrating lipidomics with other omics, such as metabolomics, microbiomics, proteomics, etc., would provide a powerful tool that could help researchers identify novel therapeutic targets and biomarkers.

Many studies have been conducted using different multi-omics techniques and clinical samples to discover novel biomarkers. However, clearing up the mechanism of how markers are generated and their diagnostic value, a critical factor in drug discovery, remains a challenge. Combining multiple experimental approaches and then integrating the results is a valuable strategy to generalize human cancer’s complexity from experimental models [197]. The integration of multiple omics, such as genomics, proteomics, and metabolomics, will help us understand tumours and advance antitumor drug developments [198,199,200]. In addition, numerous studies have confirmed that developing high-throughput sequencing technologies has revolutionized multi-omics research. It is expected that multi-omics applications will increase in scope with the optimization and maturity of various technologies, making it possible to develop novel biomarkers for CRC due to multi-omics research.

Although multi-omics methods have great potential, there are still limitations and challenges to overcome. The first problem is that omics methods are expensive and require specialized equipment and high-level data analysis skills. There can also be problems in the collection of data and verification of the data because of unreliable data quality, inaccurate data sources, and nonstandard sampling. Currently, there are no standard research platforms or bioinformatics methods for the processing of large-scale omics datasets. Data processing and analysis are the biggest challenges in metabolomics studies because biological organisms contain thousands of metabolites. Additionally, numerous obstacles will need to be overcome in order to translate biomarker discoveries into clinical applications for CRC. It is still difficult to evaluate the specificity and sensitivity of candidate biomarkers due to the absence of strategies and selection panels. Due to the fact that most patient data come from different laboratories, it is also difficult to validate biomarker candidates in large cohorts of patients. To establish potential diagnostic biomarkers, further validation may be obtained through meta-analysis. Another obstacle to overcome is the heterogeneity of the patient population and their sporadic cancers. By performing advanced MS at single-cell resolution, this problem may be tackled by understanding the biological and molecular heterogeneity of disease states.

## Figures and Tables

**Figure 1 cancers-14-05545-f001:**
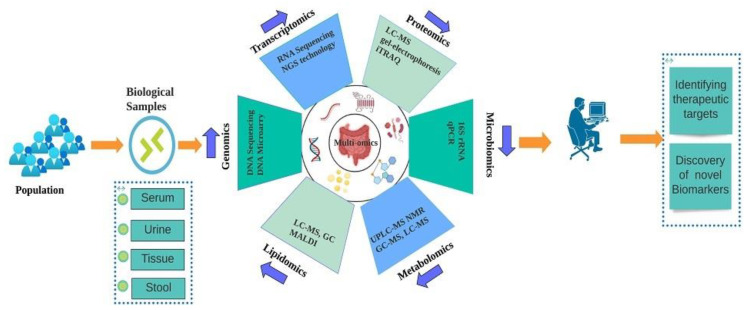
Graphical representation of different multi-omics-based approaches in discovering novel CRC biomarkers and therapeutic targets.

**Table 1 cancers-14-05545-t001:** Potential multi-omics base Genomics biomarkers in CRC.

Biomarker	Sample Type	Change	Application	References
CBX8, CD96	datasets	downregulated	diagnostic	[54]
MTUS1	tissue	downregulated	diagnostic and prognostic	[55]
SDC2, NDRG4	stool	upregulated	Screening	[56]
SOX21	stool	upregulated	diagnostic	[57]
BDNF, PTGS2, GSK3B and CTNNB1	tissue	upregulated	prognostic and diagnostic	[33]
HPGD	tissue	downregulated	prognostic and diagnostic	[33]
YWHAB, MCM4, and FBXO46	datasets	overexpress	prognostic	[58]
DPP72	datasets	lower expression	prognostic	[58]
SDC2, TFPI2	stool	hypermethylated	screening	[59]
SNORD15B, SNORA5C	tissue	upregulated	diagnostic and prognostic	[60]
GALR1	tissue	hypermethylation	screening	[61]
LRRC19	datasets	downregulated	prognosis	[62]
KRAS, BRAF, PIK3CA	tissue	mutation	detection	[63]

**Table 2 cancers-14-05545-t002:** Potential multi-omics base transcriptomics biomarkers in CRC.

Biomarker	Sample Type	Change	Application	References
miR-92a, miR-21	serum	upregulated	diagnostic and prognostic	[91]
hsa_circ_0000567	CRC tissue and cell lines	downregulated	diagnostic	[92]
hsa-circ-0006282	plasma	upregulated	Diagnostic	[93]
hsa_circ_000592, hsa_circ_0001900 and hsa_circ_0001178	plasma	upregulated	diagnostic	[94]
miR-129-1-3p mmiR-566	urine	upregulated	detection	[95]
GPR55	CRC tissue and cell lines	downregulated	prognostic	[96]
miR-1290	plasma	upregulated	prognostic	[97]
miR-320d	plasma	downregulated	diagnostic	[98]
miR-103a-3p, miR-127-3p, miR-17-5p, miR151a5p, miR-181a-5p, miR-18a-5p and miR-18b-5p	plasma	upregulated	diagnostic	[99]
CCAT2, CCAT1, H19, MALAT1, MEG3, HULC, HOTAIR, PCAT1, PTENP1 and TUSC7	stool	upregulated	detection	[100]
miR-214, miR-199a-3p, miR-196a, miR-106a, miR-183, miR-134, miR-92a, miR-96, miR-20a, miR-21, miR-17, miR-7.	stool	upregulated	screening	[101]
miR-138, miR-143, miR-29b, miR-9, miR-146a, miR-127-5p, miR-938, miR-222.	stool	downregulated	screening	[101]

**Table 3 cancers-14-05545-t003:** Potential multi-omics based proteomics biomarkers in CRC.

Biomarker	Sample Type	Change	Application	References
CHD 9	tissue	upregulated	prognostic	[117]
ACTBL2	tissue	upregulated	diagnostic	[102]
CDK3, CDK5, and CDK8	tissue	upregulated	diagnostic	[118]
STK4 or MST1	serum	downregulated	detection	[119]
MRC1 and S100A90	serum	upregulated	diagnostic	[114]
CEACAM-7	tissue	downregulated	predictive	[120]
CEA	plasma	upregulated	predictive and prognostic	[121]
SPG20 and STK31	blood	upregulated	diagnostic	[122]
TPM3	tissue/plasma	upregulated	detection	[123]
FJX1	serum	upregulated	prognostic and diagnostic	[124]
NOP14	datasets	upregulated	Prognosis	[125]
SPARCL1	datasets	Downregulated	diagnosis	[126]

**Table 4 cancers-14-05545-t004:** Potential multi-omics base microbiomics and metabolomics biomarkers in CRC.

Biomarker	Sample	Change	Application	References
*F.nucleatum, P. anaerobius* and *P. Micra*	stool	increase	detection	[155]
P. micra, Streptococcus anginosus	stool	increase	diagnosis	[156]
P. Micra F. nucleatum	stool	increase	diagnosis	[157]
norvaline and myristic acid	stool	upregulated	diagnosis	[158]
menaquinone-10	stool	upregulated	diagnosis	[159]
F. nucleatum	stool	upregulated	detection	[160]
Oleic acid	stool	Upregulated	screening	[161]
Succinate, Butyrate, Lactate, Glutamate, and Alanine.	tumour tissue/feces	Upregulated (excluding Butyrate downregulated)	detection	[152]
Cholesteryl esters, Sphingomyelins	stool	Upregulated	diagnosis	[134]
Fusobacterium, Parvimonas and Staphylococcus	stool	increase	diagnosis	[134]
Pyruvic acid, lysine, glycolic acid, fumaric acid, ornithine	blood	upregulated	detection	[162]
tryptophan, Palmitoleic acid, lysine, 3hydroxyisovaleric acid	blood	decrease	detection	[162]
octadecanoic acid, citric acid, hexadecanoic acid, and propanoic acid-2-methyl-1-(1,1-dimethylethyl)-2-methyl-1,3-propanediyl este	urine	downregulated	screening	[163]
Hydroxyproline dipeptide, tyrosine, tryptophan, pseudouridine, glucuronic acid, glycine, histidine, glucose, 5-oxoproline, threonic acid, and isocitric acid	urine	upregulated	screening	[163]

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
