# Peer review of "Multi-Omics Approaches in Colorectal Cancer Screening and Diagnosis, Recent Updates and Future Perspectives"

_cancers, 2022, doi:10.3390/cancers14225545_

Round 1
Reviewer 1 Report
Please correct grammar, spelling, punctuation mistakes.
Author Response
Response to Reviewer 1 Comments
Dear Reviewer,
First of all, we are very thankful for your positive and constructive comments and suggestion. We appreciate the time and effort that you dedicated to providing valuable feedback on our manuscript. Changes have been added to reflect the suggestions provided by the reviewer.
Point : Please correct grammar, spelling, punctuation mistakes.
Answer : Indeed we found the suggestion meaningful. For copy-editing and English proofing, a native English speaking professor checked the revised version of our manuscript in order to improve the quality of communication. Next, a Professor and domain expert who has several publications in reputed international journals rechecked our document’s revised version, structure, and language. Now we believe manuscript is free of spelling, punctuation, and grammatical errors. The revision has improved the clarity, flow, consistency, and conciseness of text with restructured sentences to improve their readability.

Reviewer 2 Report
Figure 1 is cut off--cant see it
Author Response
Response to Reviewer 2 Comments
Dear Reviewer,
First of all, we are very thankful for positive and constructive comments and suggestion. We appreciate the time and effort that you dedicated to provide a valuable feedback on our manuscript. Changes have been added to reflect the suggestions provided by the reviewer.
Point: Figure 1 is cut off-cant see it
Answer: Thank you for pointing this out. according to your suggestion, The Figure has been centralized on the page as you can see between lines 103-104.

Reviewer 3 Report
The authors touch upon a very popular topic of colorectal cancer screening and treatment. The topic is important in the clinical practice of a doctor. Unfortunately, there are some points that need to be discussed. References to sources are not given everywhere
Minor points:
lines:
42 - please change the word "algorithms" to "options"
43 - What about others specific markers, that must be checked by NCCN guidelines (KRAS, MSI-H)?
53 - Please check the sentences to correct position dot and comma
59 - change "technique" to "methods"
74 - reference?
Figure 1 - localization of the figure shout be in the center of the page, unfortunately right part is outside of the page
87 - "Gemomics of..." what?
89-92 - references?
93-95 what is the meaning of this sentence? please write the ccorrect in meaning sentence.
111 - reference?
142 - snapshot - change it, because this word is usually used in photo or IT area.
201-206 - references ?
285 - references?
Major:
1. paragraph 2 - write the short introduction
2. after all tables you have decoding the names of the markers, please relocate it to the Table.
3. What about single-cell rna seq in cancer? I havent find this information.
4. At the discussion, please add the information about the problems and disadvantages of using multi-omics markers (economic, lab problems and others).
Author Response
Response to Reviewer 3 Comments
Dear Reviewer,
First of all, we are very thankful for your positive and constructive comments and suggestions. We appreciate the time and effort that you dedicated to providing valuable feedback on our manuscript. Changes have been added to reflect the suggestions provided by the reviewer. Revised portions are marked in red on the paper.
Minor points:
Point 1: please change the word "algorithms" to "options"
Answer 1: the word "algorithms" to "options" has been changed as you can see in line 43
Point 2: What about other specific markers, that must be checked by NCCN guidelines (KRAS, MSI-H)?
Answer 2: thank you for your comments and for pointing out information about (KRAS, and MSI-H) that has been added in the revised manuscript as you can see from lines 45-49. Furthermore, information about KRAS, BRAF, PIK3CA, and biomarkers has also been added in table 1 last sentence.
Point 3: Please check the sentences to correct the position dot and comma
Answer 3:the sentence has been checked to correct position and comma lines 58-59.
Point 4: change "technique" to "methods"
Answer 4: the word technology has been changed to methods see line 64.
Point 5: reference?
Answer 5:refrence has been added can see it in line 83.
Point 6: Figure 1 - localization of the figure should be in the center of the page, unfortunately, right part is outside of the page
Answer 6:The Figure has been centralized on the page, as you can see between lines 103-104.
Point 7: "Genomics of..." what?
Answer 7:answer has been added at line 106.
Point 8: references?
Answer 8:refrences has been added as you can see on line 108-111.
Point 9: what is the meaning of this sentence? please write the correct meaning sentence.
Answer 9: Considering the Reviewer's comments, the sentence has been rewritten in a correct and meaningful way see lines 114-115.
Point 10: reference?
Answer 10:refrence has been added can see in line 132.
Point 11: snapshot - change it, because this word is usually used in a photo or IT area.
Answer 11:the word snapshot has been changed can be seen at line 163.
Point 12: references?
Answer 12:refrences has been already added to the table can be seen in the table between line 244-245.
Point 13: references?
Answer13:refrence has been added can see in line 329.
Major points:
Point 1: paragraph 2 - write a short introduction
Answer 1:Considering the Reviewer’s suggestion short introduction has been added as you can see from lines 75-79 and 83-93.
Point 2: after all tables, you have decoded the names of the markers, please relocate them to the Table.
Answer 2:the markers' names have been relocated to the tables and the abbreviations have been written on page number 15-16. If you have any query I am pleased to hear from you.
Point 3: What about single-cell rna seq in cancer? I haven't found this information.
Answer 3:Thank you for pointing this out. the comment is valuable to our manuscript. Therefore, we have added a paragraph about single-cell rna seq in cancer as you can see from lines 221-241.
Point 4: At the discussion, please add information about the problems and disadvantages of using multi-omics markers (economic, lab problems, and others).
Answer 4:Considering the Reviewer’s suggestion, We have added information in the last paragraph of the manuscript from lines 571-587.

Round 2
Reviewer 3 Report
Dear authors, thank you for your corrections.
Everything is okey, but please check the reference 18 - it doesn't have information about CRC, but the sentence of this reference is about CRC.
Author Response
Response to Reviewer 3 Comments
Dear Reviewer,
First of all, we are very thankful again for your positive and constructive comments and suggestions. We appreciate the time and effort that you dedicated to providing valuable feedback on our manuscript. Changes have been added to reflect the suggestions provided by the reviewer. Revised portions are marked in red on the paper.
Point: Everything is okey, but please check the reference 18 - it doesn't have information about CRC, but the sentence of this reference is about CRC.
Answer: Thank you for pointing this out. the reference has been checked. considering the Reviewer's suggestion, the reference has been added as can see in line 83.
